# Ancient Evolutionary History of Human Papillomavirus Type 16, 18 and 58 Variants Prevalent Exclusively in Japan

**DOI:** 10.3390/v14030464

**Published:** 2022-02-24

**Authors:** Kohsei Tanaka, Gota Kogure, Mamiko Onuki, Koji Matsumoto, Takashi Iwata, Daisuke Aoki, Iwao Kukimoto

**Affiliations:** 1Pathogen Genomics Center, National Institute of Infectious Diseases, Tokyo 208-0011, Japan; kohsei@nih.go.jp (K.T.); gota@nih.go.jp (G.K.); 2Department of Obstetrics and Gynecology, Keio University School of Medicine, Tokyo 160-0016, Japan; iwatatakashi@1995.jukuin.keio.ac.jp (T.I.); aoki@z7.keio.jp (D.A.); 3Department of Obstetrics and Gynecology, Showa University School of Medicine, Tokyo 142-8666, Japan; monuki@med.showa-u.ac.jp (M.O.); matsumok@mui.biglobe.ne.jp (K.M.)

**Keywords:** human papillomavirus, Bayesian phylogenetics, most recent common ancestor, upper paleolithic era, Japanese archipelago

## Abstract

Human papillomavirus (HPV) is a sexually transmitted virus with an approximately 8-kilo base DNA genome, which establishes long-term persistent infection in anogenital tissues. High levels of genetic variations, including viral genotypes and intra-type variants, have been described for HPV genomes, together with geographical differences in the distribution of genotypes and variants. Here, by employing a maximum likelihood method, we performed phylogenetic analyses of the complete genome sequences of HPV16, HPV18 and HPV58 available from GenBank (*n* = 627, 146 and 157, respectively). We found several characteristic clusters that exclusively contain HPV genomes from Japan: two for HPV16 (sublineages A4 and A5), one for HPV18 (sublineage A1) and two for HPV58 (sublineages A1 and A2). Bayesian phylogenetic analyses of concatenated viral gene sequences showed that divergence of the most recent common ancestor of these Japan-specific clades was estimated to have occurred ~98,000 years before present (YBP) for HPV16 A4, ~39,000 YBP for HPV16 A5, ~38,000 YBP for HPV18 A1, ~26,000 for HPV58 A1 and ~25,000 YBP for HPV58 A2. This estimated timeframe for the divergence of the Japan-specific clades suggests that the introduction of these HPV variants into the Japanese archipelago dates back to at least ~25,000 YBP and provides a scenario of virus co-migration with ancestral Japanese populations from continental Asia during the Upper Paleolithic period.

## 1. Introduction

Human papillomaviruses (HPVs) are a family of small non-enveloped viruses having a circular double-stranded DNA genome of approximately 8000 bp [1]. More than 200 genotypes have been identified based on a >10% difference in the *L1* capsid gene sequence [2]. HPVs infect basal epithelial cells in either cutaneous or mucosal tissues, and around 60 genotypes are phylogenetically clustered and recognized as mucosa-tropic genotypes [3]. Such HPVs are considered to be sexually transmitted, and at least 15 types are causatively associated with the development of human malignancies, including cervical, other anogenital, and head-and-neck cancers [4]. HPV16 is the most prevalent type detected in cervical cancer worldwide, followed by HPV18 [5]. Although HPV58 is the seventh most common type of cervical cancer across the world, it is the third most dominant type in East Asia, due to geographical variations of HPV type distribution.

Despite the existence of a large number of genotypes, HPV genome sequences are considered to be highly stable compared to RNA viruses because HPV genome replication relies completely on the high-fidelity DNA polymerases of the host cell [6]. Within individual genotypes, however, the HPV genome harbors another level of genetic complexity, called intra-type variants, which generally show less than 10% differences in the complete viral genome sequence. In this context, 1.0–10% and 0.5–1.0% nucleotide differences are defined as variant lineages and sublineages, respectively [7]. Indeed, a recent genomics study of HPV16 documented more than 2000 slightly different viral genomes from clinical samples [8]. Such high levels of minor genetic variations are most likely the results of viral genome editing or mutagenesis by host-cell cytosine deaminases of the APOBEC3 family during the long evolutionary history of virus-host interaction [9,10,11]. 

The time scale of PV evolution over millions of years of co-evolution with their hosts is largely inferred from a model of virus-host co-divergence [12,13,14]. This is because vertebrate PVs exhibit strict host specificity for infection and the molecular phylogeny of PV genomic sequences shows a concordance with that of the host species, despite some incongruences being present [15]. Recently, regarding HPV genomic diversity, several studies have provided intriguing scenarios for the evolutionary history of HPVs. Firstly, by dating the divergence times of HPV16 variant lineages with a Bayesian statistics framework, Pimenoff et al. showed that HPV16 A and B/C/D variant lineages split apart ~500,000 years before present (YBP), which largely predates the birth of *Homo sapiens* (~200,000 YBP) and coincides with the timing of the split between Neanderthals and *Homo sapiens* [16]. Considering the geographic distribution of extant HPV16 variants (e.g., lineage A variants are exclusively prevalent in Eurasia, whereas lineage B variants are dominant only in Africa), these data support the Hominin-host-switch model, in which sexual transmission of the HPV16 A lineage from archaic to modern human populations occurred in Eurasia after out-of-Africa migration of *Homo sapiens* ~90,000 YBP. To precisely annotate the divergence times of HPV16 variants, a subsequent study by Chen et al. established a molecular clock model inferred from PV sequences of non-human primates, which also estimated the first divergence of HPV16 variants to have occurred around 500,000 YBP [17]. Similar evolutionary timeframes of the radical divergence of major variant lineages were also reported for other HPV types [17,18]. Finally, recent studies implicated intra-host divergence following initial adaptation to specific niches, such as cutaneous or mucosal tissues, as a primary evolutionary pathway of human and non-human primate PVs, rather than strict virus-host co-divergence [18,19,20]. 

Although virus transmission from archaic humans is difficult to prove due to the lack of direct detection of HPV DNA in archaeological remains, virus co-migration with modern human populations after out-of-Africa migration seems to have had a significant impact on the present-day distributions of HPV genotypes and variants across the world [16,17]. Previously, we identified a genetic cluster of HPV16 variants that are exclusively prevalent in Japan, which we named sublineage A5 [21]. In the current study, we have comprehensively surveyed specific variants for HPV16, HPV18 and HPV58, and found similar Japan-specific clusters for all three genotypes. By dating the divergence times of these Japan-specific variants with a Bayesian Markov Chain Monte Carlo (MCMC) method, we discuss the evolutionary histories of these viruses based on the migration history of ancient Japanese ancestors.

## 2. Materials and Methods

### 2.1. Viral Genome Sequences

The complete genome sequences of HPV16 (*n* = 627), HPV18 (*n* = 146) and HPV58 (*n* = 157) genomes were retrieved from GenBank using the search terms “human papillomavirus type 16” [Primary Organism] OR “human papillomavirus type 18” [Primary Organism] OR “human papillomavirus type 58” [Primary Organism] AND complete genome [TITLE] (as of 31 August 2021) for phylogenetic analyses. Additionally, using next-generation sequencing approaches, as described previously [21], 49 complete genomic sequences of HPV16 were newly determined from cervical cancer and precancer specimens from women in Japan (the accession numbers are LC644164 to LC644191 and LC647437 to LC647457) and are included in this study. For comparison with HPV16 variants isolated from the United States, HPV16 sequences belonging to the sublineage A4 (*n* = 42) were extracted from a set of viral genome sequences reported by Mirabello et al. [8]; the list of accession numbers is shown in Appendix A.

### 2.2. Maximum Likelihood Phylogenetic Analyses

For each genotype, virus complete genomic sequences including representative sequences for variant lineages/sublineages were aligned against each other using MAFFT v7.309 with default parameters. Maximum likelihood trees were inferred using RAxML v8.2.9 under the general time-reversible nucleotide model with gamma-distributed rate heterogeneity and invariant sites (GTR + G + I), employing 1000 bootstrap replicates. Phylogenetic trees were visualized using FigTree v1.4.2 (http://tree.bio.ed.ac.uk/software/figtree/, accessed on 31 August 2021).

### 2.3. Bayesian Phylogenetic Analyses

A Bayesian MCMC framework implemented by BEAST v2.6.5 was used to estimate the divergence time of the most recent common ancestor (MRCA) for HPV16/18/58 variants. Appropriate clock and tree models were determined for our datasets by the pass sampling method using the Path sampler in the BEAST2 packages, leading to selection of relaxed lognormal molecular clock and coalescent Bayesian skyline models. The concatenated nucleotide sequences of six open-reading frames (*E6*, *E7*, *E1*, *E2*, *L2* and *L1*) and a previously reported HPV16 evolutionary rate of 1.84 × 10^−8^ substitutions/site/year [16] were used for all MCMC analyses. The most appropriate substitution models were determined by the best-fit model approach of jModelTest v2.1; the selected models were GTR + I + G for HPV16, TPM1uf + I for HPV18 and TVM + I + G for HPV58. The MCMC analysis was run for 100,000,000 chains, with sampling every 10,000 generations. After the first 10% of the chain was omitted, effective sample sizes (ESS) were monitored by Tracer v1.7.1 and ESS greater than 200 were accepted. Maximum clade credibility trees were constructed using Tree Annotator v2.6.0, and the MCMC phylogenetic trees were visualized using FigTree v1.4.2.

## 3. Results

### 3.1. Japan-Specific Clusters of HPV16/18/58 Genomes

Thus far, we have determined a number of complete genomic sequences of HPV16 (*n* = 172), HPV18 (*n* = 21) and HPV58 (*n* = 57) from cervical swab samples of Japanese women [21,22,23]. To compare phylogenetic relationships of these Japanese isolates with HPV isolates reported worldwide, we used a maximum likelihood framework to construct phylogenetic trees of whole viral genomes. 

The phylogenetic tree of a total of 627 complete genomes of HPV16 showed that the Japanese isolates were distributed across the lineages A, C and D (Figure 1), but most of them belonged to the lineage A (161/172, 93.6%), in particular the sublineage A4 (88/172, 51.2%), which had originally been designated as the Asian lineage. Some of the Japanese isolates were also clustered into the clade of the sublineage A5 (28/172, 16.3%) and, as we previously reported [21], this A5 cluster contained HPV16 genomes exclusively from Japan. Regarding the A4 variants from Japan, about one-half (41/88, 46.6%) were clearly separated from the clade that included the reference A4 genome near the root of the tree. To determine whether this A4 cluster was unique to the HPV16 genomes from Japan, we accessed additional A4 genome sequences (*n* = 42) from a recent study of HPV16 in the United States [8], and performed phylogenetic analysis with these. It is shown in Appendix A, with one exception (accession number, MG848021); none of the US isolates were grouped into this cluster, indicating regional exclusivity or specificity of this A4 cluster harboring the Japanese isolates.

The phylogenetic tree of a total of 146 complete genome sequences of HPV18 revealed that most of the HPV18 isolates from Japan belonged to the sublineage A1 (19/21, 90.5%) (Figure 2). Among the A1 genomes from Japan, some (7/21, 33.3%) formed a cluster distinct from other A1 genomes reported across the world, suggesting the presence of Japan-specific HPV18 variants.

The phylogenetic tree of a total of 157 complete genome sequences of HPV58 showed that these genomes from Japan were widely distributed across the sublineages A1, A2 and A3 (Figure 3). Again, a cluster of the sublineage A1 that only contained the Japanese isolates (13/57, 22.8%), and a smaller cluster of the sublineage A2 that was enriched for the HPV58 genomes from Japan (7/57, 12.3%), were found for HPV58, indicating that distinct exclusively Japanese clades were present for all three genotypes.

### 3.2. Estimation of the Divergence Time of Japan-Specific HPV16/18/58 Variants

To estimate the evolutionary timeframe of the emergence of the Japan-specific HPV16/18/58 genomes, we employed a Bayesian phylogenetic framework for subsequent analyses. The relaxed lognormal molecular clock and coalescent Bayesian skyline models, which were validated for HPV genome evolution in recent studies [16,17], were used with the concatenated virus gene sequences of *E6*, *E7*, *E1*, *E2*, *L2* and *L1*. 

The time-scaled maximum clade credibility tree of HPV16 showed a similar tree topology to the maximum likelihood tree (Figure 4). The deepest divergence between the lineage A and B/C/D was calculated to be 510,349 YBP (95% highest posterior density [HPD] interval: 260,852–801,328), which was consistent with the results of previous studies, which were 461,000 YBP [16] and 488,900 YBP [17]. As shown in Table 1, the divergence times of the MRCAs were estimated to be 39,279 YBP (95% HPD interval: 21,572–63,467) for the Japan-specific A5 variants and 97,091 YBP (95% HPD interval: 61,272–136,486) for the Japan-specific A4 variants. 

The HPV16 sequence dataset of Pimenoff et al. [16] included seven HPV16 isolates from Japan, which was used to confirm our divergence time estimates. The Bayesian phylogenetic analysis of this dataset reconstructed a similar timeframe for HPV16 variant evolution as in the previous study (Appendix A). The divergence time of the MRCA for two Japanese A5 isolates (AB818687 and AB818688) and a Thai isolate (FJ610151) was estimated to be 39,681 YBP (95% HPD interval: 20,423–64,452) (Table 1). One Japanese isolate (AB818691), belonging to the Japan-specific A4 cluster in our current study, diverged from the other A4 variants at 103,650 YBP (95% HPD interval: 66,397–144,197).

For HPV18, an MCMC tree with a similar topology to the maximum likelihood tree was also reconstructed (Figure 5). The deepest divergence between the lineages A and B/C/D was estimated to be 543,988 YBP (95% HPD interval: 339,460–775,927), consistent with 552,100 YBP result of a previous study [17]. The divergence time of the MRCA of the Japan-specific A1 variants was estimated to be 37,702 YBP (95% HPD interval: 26,348–50,678) (Table 1). 

For HPV58, the MCMC tree also showed a topology consistent with the maximum likelihood tree (Figure 6). The deepest divergence between the lineages A and B/C/D was estimated to be 412,795 YBP (95% HPD interval: 247,936–603,724), also consistent with the result of Chen et al., which was 478,600 YBP [18]. The divergence times of the MRCAs were estimated to be 25,879 YBP (95% HPD interval: 17,403–30,208) for the Japan-specific A1 variants and 25,032 YBP (95% HPD interval: 16,461–29,031) for the Japan-specific A2 variants (Table 1).

## 4. Discussion

In this study, we explored the phylogenetic relationships discernable from the complete sequences of HPV16/18/58 genomes reported worldwide. We found characteristic genetic clusters that exclusively contain HPV variants from Japan in each of these three genotypes. These clusters belonged to the sublineages A4 and A5 for HPV16, the sublineage A1 for HPV18, and the sublineages A1 and A2 for HPV58. We then performed the Bayesian phylogenetic analyses to infer the evolutionary timeframe for the emergence of these Japan-specific variants, and showed that except for HPV16 A4 the divergence time of the MRCAs of these variants dated back to between 40,000–25,000 YBP, corresponding to the Upper Paleolithic era. How does this estimated timeframe explain the emergence of the Japan-specific HPV variants in relation to the history of the ancient Japanese?

The Japanese archipelago is located east of the Asian continent and has seen human activity since at least 30,000 YBP. Regarding the origin of the current Japanese, two prehistoric overseas migrations are considered to have contributed to the formation of the ancestral Japanese population (Figure 7) [24]. One of these was the first waves of migration from mainland Asia, which are supposed to have occurred during the late Pleistocene between 40,000–30,000 YBP, based on archaeological evidence such as stone tools and the oldest human remains dating back to ~36,500 YBP [25]. This Upper Paleolithic period was followed by the Jomon period that lasted from 16,500 to 3000 YBP, lived by the Jomon people with a hunter-gatherer lifestyle. The second migration events from Northeast Asia occurred after ~3000 YBP in the Holocene, together with the introduction of rice farming into the Japanese archipelago, which marks the start of the Yayoi period. Recent genome-wide single-nucleotide polymorphisms studies of the current and ancient Japanese [26,27,28,29] support the dual structure model, in which the Yayoi migrants interbred with the indigenous Jomon people [30,31], as a model for the origin of the present-day Japanese. 

Considering these population histories, our estimated divergence times of the MRCAs of the Japan-specific HPV16/18/58 genomes imply that these variants were introduced into the Japanese archipelago, most probably with the first migration at 40,000–25,000 YBP (Figure 7). Although the population continuity between the Paleolithic and Jomon periods remains elusive, we speculate that these HPV variants were spread via sexual transmission within the prehistoric Japanese population in the Upper Paleolithic era and subsequently passed on to the Jomon people, a significant source of genetic ancestry of the current Japanese.

In contrast, the deep divergence of the Japan-specific HPV16 A4 variants from other A4 variants (~97,000 YBP) indicates that these variants were generated much earlier than other Japan-specific variants. Recent studies investigating the divergence time of HPV16 variants suggest that the major split of A4 variants from the A1/2/3 variants occurred ~100,000 YBP [16] and ~200,000 YBP [17]. This suggests transmission of A4 variants from archaic hominins, such as Denisovans, to *Homo sapiens* in East Eurasia, which explains the exclusive prevalence of A4 variants in this region today. The deep divergence observed for the Japan-specific A4 variants in our study may also reflect such viral transmission from archaic hominins to Japanese ancestors in continental Asia, followed by co-migration of these variants into the Japanese archipelago. A recent study on ancient DNA from Jomon individuals indicated that the Jomon lineage diverged before the diversification of present-day East Eurasian populations, suggesting that the Jomon is one of the oldest lineages in East Eurasia [28,29]. Thus, the ancestral population of the Jomon lineage in East Eurasia might have introduced the Japan-specific A4 variants into the Japanese archipelago, without leaving any trace of such variants in the place from which they departed. Alternatively, because the HPV16 genomes analyzed in our study included only seven isolates from China and five from Thailand, HPV16 variants that are closely related to the Japan-specific A4 variants might remain undiscovered in East/Southeast Asia. Such yet-unexplored variants may fill the gap of the evolutionary timeframe of the Japan-specific A4 variants and provide further insight into the evolution of HPV16 variants.

Our scenario that particular HPV variants have been continuously present within the Japanese population for at least ~25,000 years might be surprising, but also conceivable given that sexual transmission is one of the best strategies for a virus to sustain its infection cycles from generation to generation, and that the Japanese archipelago is geographically isolated and experienced no large-scale migration events after the Yayoi period. Another important factor that might facilitate the maintenance of specific HPV variants within a population is viral adaptation to that particular population. In this regard, the Japan-specific HPV16 A5 genomes encode a characteristic E2 protein, A105T, not seen in other variant genomes [21]. The E2 protein is a viral replication/transcription factor and is required for stable maintenance of virus genomes through cell division [32]. E2 expression in cervical basal cells is considered to be critical for long-term persistent infection [33], and its immune recognition by cytotoxic T-lymphocytes (CTL) likely contributes to clearing HPV infections [34,35]. Thus, the A105T variant may assist immune evasion by changing the CTL epitopes in the E2 protein. For instance, if these altered CTL epitopes are not optimal, particularly in Japanese people, this could facilitate long-term viral persistence. Because E2 amino acid changes are also observed for the Japan-specific HPV58 A1 variants (i.e., S275N), further studies should be focused on the immunological properties of this viral protein.

## 5. Conclusions

This study has revealed the ancient history of Japan-specific HPV variants, strongly suggesting virus co-migration with prehistoric Japanese ancestors into the Japanese archipelago. Given an extremely slow rate of HPV evolution, it can be envisioned that other regions of the world have also kept their own HPV variants within their populations, and that such HPV variants may provide novel insights into a yet-to-be-defined history of human populations.

## Figures and Tables

**Figure 1 viruses-14-00464-f001:**
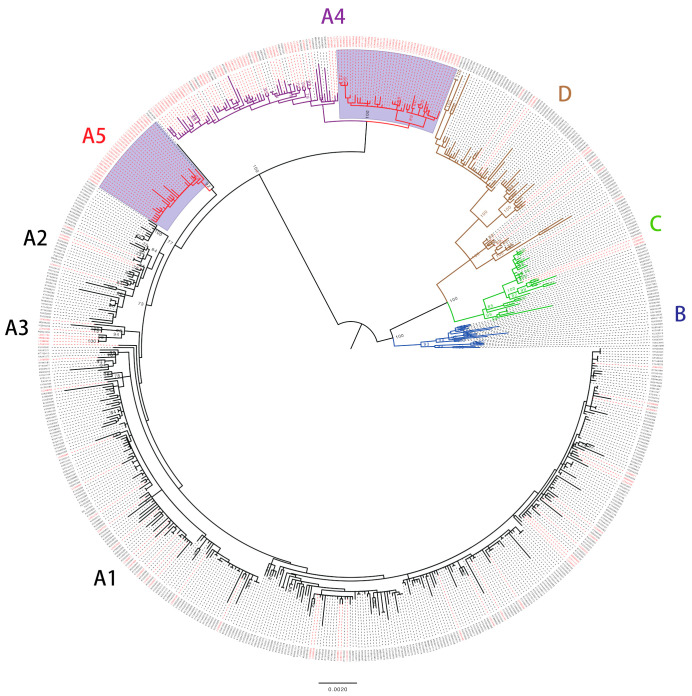
Maximum likelihood phylogenetic tree of complete HPV16 genomes available from GenBank (https://www.ncbi.nlm.nih.gov/genbank/, accessed on 31 August 2021). A total of 627 sequences were analyzed in RAxML with 1000 bootstrap replicates. Bootstrap values >70% are displayed. Red taxa indicate 172 sequences from Japan. Purple area marks the Japan-specific cluster of the sublineage A4 and the cluster of the sublineage A5. Scale bar, nucleotide substitutions per site.

**Figure 2 viruses-14-00464-f002:**
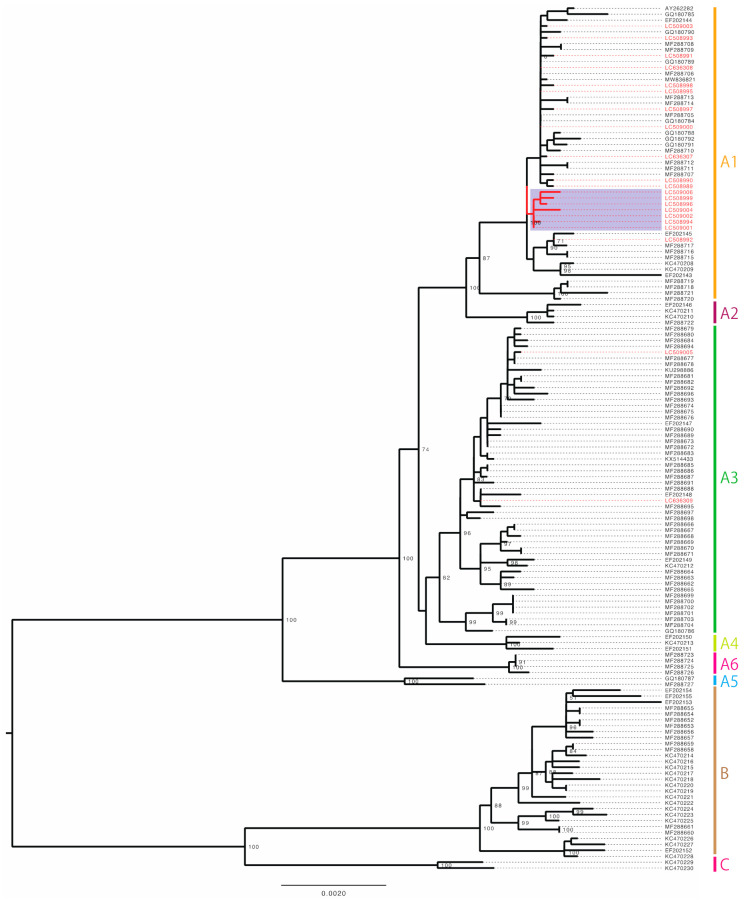
Maximum likelihood phylogenetic tree of complete HPV18 genomes available from GenBank (https://www.ncbi.nlm.nih.gov/genbank/, accessed on 31 August 2021). A total of 146 sequences were analyzed in RAxML with 1000 bootstrap replicates. Bootstrap values >70% are displayed. Red taxa indicate 21 sequences from Japan. Purple area marks the Japan-specific cluster of the sublineage A1. Scale bar, nucleotide substitutions per site.

**Figure 3 viruses-14-00464-f003:**
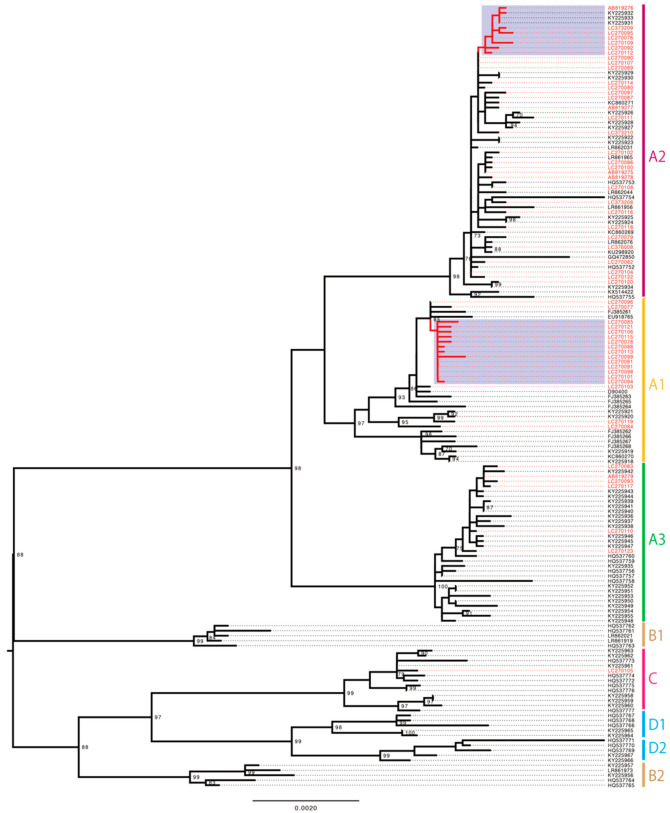
Maximum likelihood phylogenetic tree of complete HPV58 genomes available from GenBank (https://www.ncbi.nlm.nih.gov/genbank/, accessed date: 31 August 2021). A total of 157 sequences were analyzed in RAxML with 1000 bootstrap replicates. Bootstrap values >70% are displayed. Red taxa indicate 57 sequences from Japan. Purple area marks the Japan-specific cluster of the sublineages A1 and A2. Scale bar, nucleotide substitutions per site.

**Figure 4 viruses-14-00464-f004:**
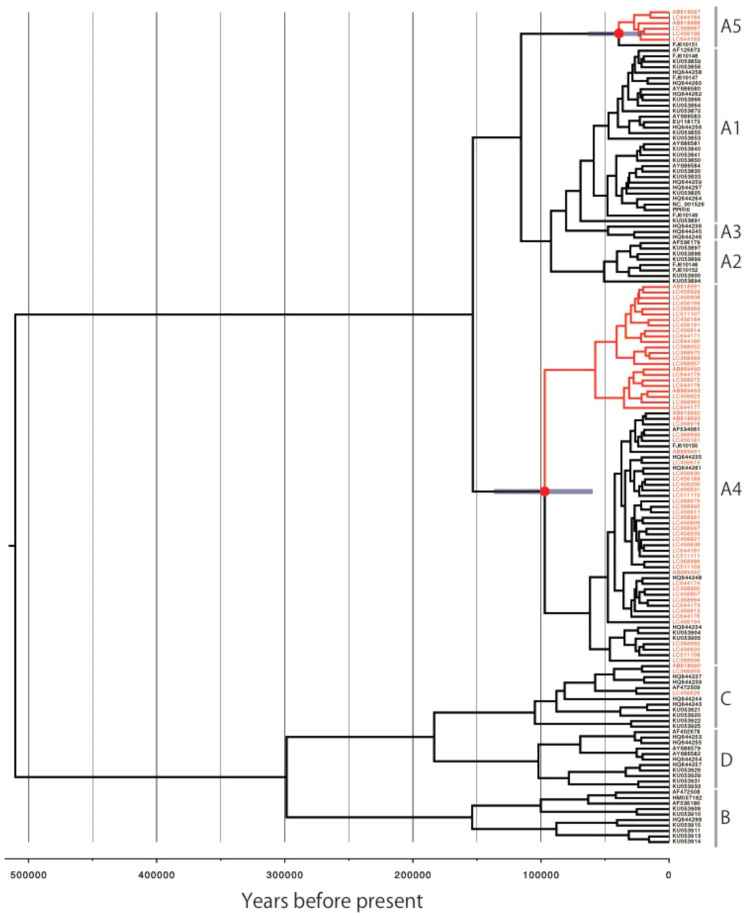
Bayesian MCMC phylogenetic tree of HPV16. The concatenated *E6*, *E7*, *E1*, *E2*, *L2* and *L1* sequences from 152 HPV16 genomes were analyzed in BEAST2. Time-scaled maximum-clade credibility tree is shown. Red node indicates the position of the most recent common ancestor for the Japan-specific HPV16 variants. Blue bar indicates 95% highest posterior density interval.

**Figure 5 viruses-14-00464-f005:**
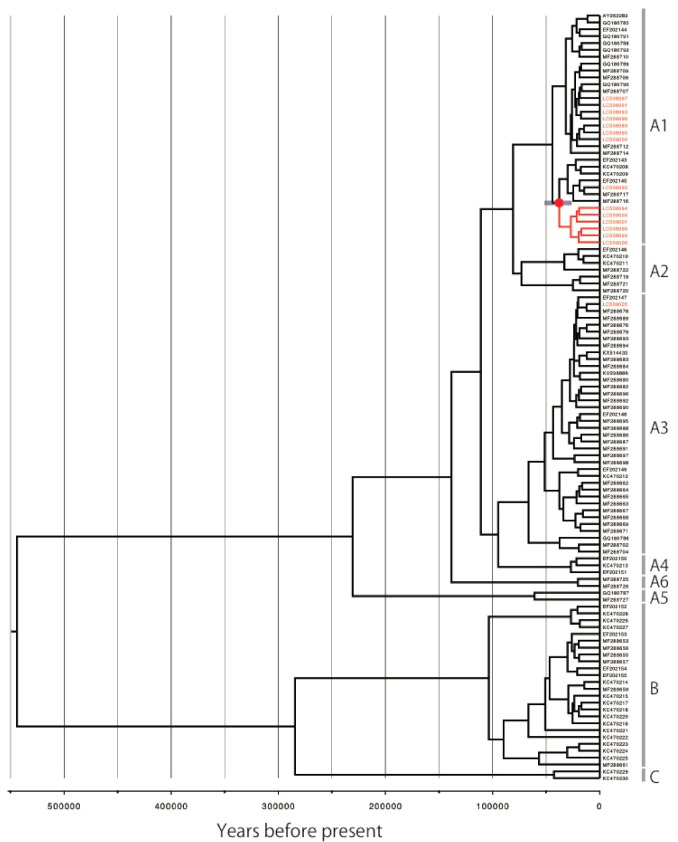
Bayesian MCMC phylogenetic tree of HPV18. The concatenated *E6*, *E7*, *E1*, *E2*, *L2* and *L1* sequences from 112 HPV18 genomes were analyzed in BEAST2. Time-scaled maximum-clade credibility tree is shown. Red node indicates the position of the most recent common ancestor for the Japan-specific HPV18 variants. Blue bar indicates 95% highest posterior density interval.

**Figure 6 viruses-14-00464-f006:**
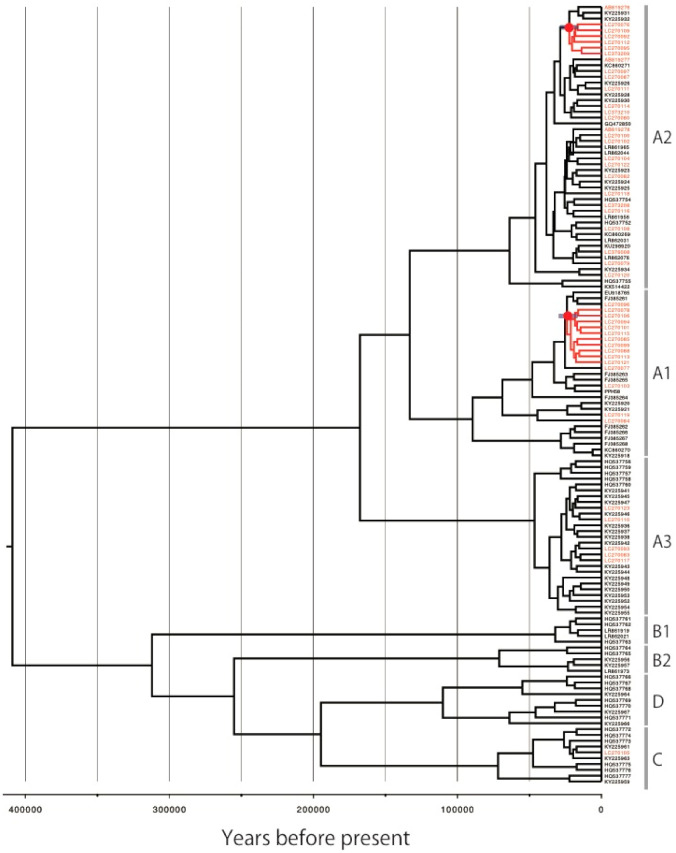
Bayesian MCMC phylogenetic tree of HPV58. The concatenated *E6*, *E7*, *E1*, *E2*, *L2* and *L1* sequences from 134 HPV58 genomes were analyzed in BEAST2. Time-scaled maximum-clade credibility tree is shown. Red node indicates the position of the most recent common ancestor for the Japan-specific HPV58 variants. Blue bar indicates 95% highest posterior density interval.

**Figure 7 viruses-14-00464-f007:**
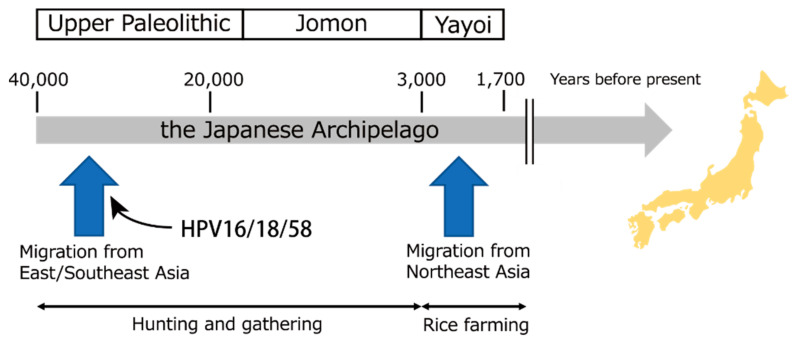
Timeframe of migration of ancestral Japanese populations into the Japanese archipelago and co-migration of Japan-specific HPV16/18/58 variants.

**Table 1 viruses-14-00464-t001:** Divergence time estimates of MRCAs of the Japan-specific HPV variants.

Type	Variant	Divergence Time (Years Ago)	95% HPD Interval
HPV16	Japan A5 vs. FJ610151	39,279	21,572–63,467
HPV16	Japan A4 vs. Other A4	97,091	61,272–136,486
HPV16 *	Japan A5 vs. FJ610151	39,681	20,423–64,452
HPV16 *	Japan A4 vs. Other A4	103,650	66,397–144,197
HPV18	Japan A1 vs. Other A1	37,702	26,348–50,678
HPV58	Japan A1 vs. Other A1	25,879	17,403–30,208
HPV58	Japan A2 vs. Other A2	25,032	16,461–29,031

*, the dataset of Pimenoff et al. (118 complete genomes) was used. Divergence time estimates are shown as mean values. HPD, highest posterior density.

## Data Availability

All HPV genome sequences used are available from the GenBank database.

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
