# Peer review of "Ancient Evolutionary History of Human Papillomavirus Type 16, 18 and 58 Variants Prevalent Exclusively in Japan"

_viruses, 2022, doi:10.3390/v14030464_

Round 1

Reviewer 1 Report

This is a straightforward phylogenetic study that examined HPV viral genome sequences from three viral genotypes, comparing viruses isolated in Japan with control genomes deposited in public databases. The authors demonstrated that the Japan-specific viral sequences appear to be restricted to the Japanese population. Estimates of the appearance and divergence for each of these viral populations correlate reasonably well with the two documented major migrations to Japan from the East Asian continental population, suggesting that these viruses arrived with the immigrant populations, and constitute a bottlenecked population that has remained isolated since that time. The manuscript is well-written and the methods appear to be appropriately employed and described. I cannot find any points of serious concern for this manuscript.

Author Response

Thank you very much for your favorable evaluation of our manuscript. We greatly appreciate your reviewing.

Reviewer 2 Report

In this manuscript, the authors have sequenced new isolates of HPV from Japan and performed Bayesian phylogenetics along with previously reported HPV sequences. They have identified Japan-specific phylogenetics tree clusters and estimated their divergence time. Further, phylogenics data was correlated with the archipelago data and suggested that the HPV co-migrated into Japan from continental Asia.   The highlights of this manuscript are (1) obtaining the complete genome sequence of49 new isolates of HPV from Japan and identifying sublineages clusters of the viruses exclusive to Japan (2) estimating divergence of the most recent common ancestor for those clusters, and (3) relating bayesian data with archipelago data. However, the weakness of this manuscript comes from defining the best fit Bayesian model and nucleotide substitution data. The authors have used previously reported data for the analysis here. Though the models and nucleotide substitution data are well defined for HPV and reported in manuscripts published in 2017, here authors could have considered to re-do the analysis using added new genome data (I assume the best fit model output might be the same but will definitely see changes in nucleotide substitution changes and that improved estimated data).     MINOR CRITIQUES.   1. Authors could have considered identifying amino acids in the coding sequence undergoing positive/ negative selection within the HPV Japan isolates.   2. Clarify divergence time provided as mean or median value.   3 Figure 2: Annotation of A5 is not clear with a gap between A5 and A4.   4. Figures 4, 5, and 6: If possible, provide Corresponding 95% highest posterior density (HPD) values of TMRCA are indicated as gray bars.   5. Figure 5: Please re-check the sidebar provided to indicate A4 is properly represented.   6. Figure 6:Please re-check the sidebar provided to indicate B is properly represented. I see that includes different branches of the tree.   7. In this manuscript, the sequences were retrieved and analyzed from GenBank is as of August 31st, 2021. If a new sequence were submitted and available now, the authors could have included them as well in this manuscript and updated the results. In addition, the Bayesian analysis models and nucleotide substitution rates applied in this manuscript were extrapolated from previous reports. Therefore, re-analysis of Bayesian analysis using recent sequences from the past 6-8 months can be performed quickly.

Author Response

We thank the reviewer for his/her valuable comments and suggestions. Our responses to the comments are listed point-by-point below.

In this manuscript, the authors have sequenced new isolates of HPV from Japan and performed Bayesian phylogenetics along with previously reported HPV sequences. They have identified Japan-specific phylogenetics tree clusters and estimated their divergence time. Further, phylogenic data was correlated with the archipelago data and suggested that the HPV co-migrated into Japan from continental Asia. The highlights of this manuscript are (1) obtaining the complete genome sequence of 49 new isolates of HPV from Japan and identifying sublineages clusters of the viruses exclusive to Japan (2) estimating divergence of the most recent common ancestor for those clusters, and (3) relating Bayesian data with archipelago data.

However, the weakness of this manuscript comes from defining the best fit Bayesian model and nucleotide substitution data. The authors have used previously reported data for the analysis here. Though the models and nucleotide substitution data are well defined for HPV and reported in manuscripts published in 2017, here authors could have considered to re-do the analysis using added new genome data (I assume the best fit model output might be the same but will definitely see changes in nucleotide substitution changes and that improved estimated data).

  • Response: For our Bayesian MCMC analysis, we selected a nucleotide substitution model and a Bayesian clock/tree model based on the analysis of our sequence datasets by jModelTest and the Path sampler in the BEAST2 packages, respectively. We have modified the Materials and Methods section to clearly indicate this issue as follows: “Appropriate clock and tree models were determined for our datasets by the pass sampling method using the Path sampler in the BEAST2 packages, leading to selection of relaxed lognormal molecular clock and coalescent Bayesian skyline models.”

MINOR CRITIQUES.  

  1. Authors could have considered identifying amino acids in the coding sequence undergoing positive/negative selection within the HPV Japan isolates.
  • Response: Thank you for your insightful suggestion. We are currently investigating the positive and negative selection of amino acid residues in the Japanese isolates and their functional impact on the HPV life cycle, and hope to report this as a separate study in the near future.

  1. Clarify divergence time provided as mean or median value.
  • Response: The divergence time estimates are expressed as mean values, and this information is added to the footprint of Table 1 as “Divergence time estimates are shown as mean values.”

3 Figure 2: Annotation of A5 is not clear with a gap between A5 and A4.

  • Response: Thank you for your valuable suggestion. In the phylogenetic tree of HPV18, a new sublineage exists between A4 and A5, recently defined as A6 (Genomics, 113 (2021), 3895–3906). We have modified Figure 2 to indicate this sublineage A6.

  1. Figures 4, 5, and 6: If possible, provide Corresponding 95% highest posterior density (HPD) values of TMRCA are indicated as gray bars.
  • Response: As suggested by the reviewer, we have shown 95% HPD of TMRCA as blue bars in modified Figures 4, 5 and 6.

  1. Figure 5: Please re-check the sidebar provided to indicate A4 is properly represented.
  • Response: We have modified Figure 5 to indicate A6 separately from A4.

  1. Figure 6: Please re-check the sidebar provided to indicate B is properly represented. I see that includes different branches of the tree.
  • Response: Thank you for your valuable suggestion. The HPV58 lineage B includes two sublineages, B1 and B2, and we have modified Figure 6 to indicate B1 and B2 separately.

  1. In this manuscript, the sequences were retrieved and analyzed from GenBank is as of August 31st, 2021. If a new sequence were submitted and available now, the authors could have included them as well in this manuscript and updated the results. In addition, the Bayesian analysis models and nucleotide substitution rates applied in this manuscript were extrapolated from previous reports. Therefore, re-analysis of Bayesian analysis using recent sequences from the past 6-8 months can be performed quickly.
  • Response: We have searched GenBank for the full-genome sequence of HPV16/18/58, but could not find any new entries since August 31, 2021.

Finally, we found that in Figure 1 annotation of A2 and A3 was bit misaligned, so we have corrected it.
